# Less Severe Lipopolysaccharide-Induced Inflammation in Conditional *mgmt*-Deleted Mice with LysM-Cre System: The Loss of DNA Repair in Macrophages

**DOI:** 10.3390/ijms241210139

**Published:** 2023-06-14

**Authors:** Wilasinee Saisorn, Pornpimol Phuengmaung, Jiraphorn Issara-Amphorn, Jiradej Makjaroen, Peerapat Visitchanakun, Kritsanawan Sae-khow, Atsadang Boonmee, Salisa Benjaskulluecha, Aleksandra Nita-Lazar, Tanapat Palaga, Asada Leelahavanichkul

**Affiliations:** 1Interdisciplinary Program of Biomedical Sciences, Graduate School, Chulalongkorn University, Bangkok 10330, Thailand; wsaisorn@gmail.com; 2Department of Microbiology, Faculty of Medicine, Chulalongkorn University, Bangkok 10330, Thailand; pphuengmaung@gmail.com (P.P.); peerapat.visitchanakun@gmail.com (P.V.); kritsanawan_29@hotmail.com (K.S.-k.); 3Center of Excellence in Translational Research in Inflammation and Immunology (CETRII), Faculty of Medicine, Chulalongkorn University, Bangkok 10330, Thailand; 4Functional Cellular Networks Section, Laboratory of Immune System Biology, National Institute of Allergy and Infectious Diseases NIH, Bethesda, MD 20892-1892, USA; jiraphorn.issara-amphorn@nih.gov (J.I.-A.); nitalazarau@niaid.nih.gov (A.N.-L.); 5Center of Excellence in Systems Biology, Research Affairs, Faculty of Medicine, Chulalongkorn University, Bangkok 10330, Thailand; jiradejmak@gmail.com; 6Department of Microbiology, Faculty of Science, Chulalongkorn University, Bangkok 10330, Thailand; atsadang88@gmail.com (A.B.); salisafaii@gmail.com (S.B.); tanapat.p@chula.ac.th (T.P.); 7Division of Nephrology, Department of Medicine, Faculty of Medicine, Chulalongkorn University, Bangkok 10330, Thailand

**Keywords:** sepsis, lipopolysaccharide, macrophages, epigenetics, *mgmt*

## Abstract

Despite the known influence of DNA methylation from lipopolysaccharide (LPS) activation, data on the O6-methylguanine-DNA methyltransferase (MGMT, a DNA suicide repair enzyme) in macrophages is still lacking. The transcriptomic profiling of epigenetic enzymes from wild-type macrophages after single and double LPS stimulation, representing acute inflammation and LPS tolerance, respectively, was performed. Small interfering RNA (siRNA) silencing of *mgmt* in the macrophage cell line (RAW264.7) and *mgmt* null (*mgmt*^flox/flox^; LysM-Cre^cre/−^) macrophages demonstrated lower secretion of TNF-α and IL-6 and lower expression of pro-inflammatory genes (*iNOS* and *IL-1β*) compared with the control. Macrophage injury after a single LPS dose and LPS tolerance was demonstrated by reduced cell viability and increased oxidative stress (dihydroethidium) compared with the activated macrophages from littermate control mice (*mgmt*^flox/flox^; LysM-Cre^−/−^). Additionally, a single LPS dose and LPS tolerance also caused mitochondrial toxicity, as indicated by reduced maximal respiratory capacity (extracellular flux analysis) in the macrophages of both *mgmt* null and control mice. However, LPS upregulated *mgmt* only in LPS-tolerant macrophages but not after the single LPS stimulation. In mice, the *mgmt* null group demonstrated lower serum TNF-α, IL-6, and IL-10 than control mice after either single or double LPS stimulation. Suppressed cytokine production resulting from an absence of *mgmt* in macrophages caused less severe LPS-induced inflammation but might worsen LPS tolerance.

## 1. Introduction

Sepsis is a potentially life-threatening condition arising in response to severe infection, regardless of the organismal causes (bacteria, viruses, fungi, and parasites) [1,2,3], partly due to the simultaneous immunological imbalance between hyperinflammation and immune exhaustion (immune paralysis) [4,5] in the same patient [6]. Sepsis-induced hyperinflammation results in severe sepsis, while immune exhaustion leads to inadequate inflammation for microbial control, causing secondary infection [7]. The blockage and enhancement of inflammation during sepsis hyperinflammation and immune exhaustion, respectively, may be helpful [8,9,10,11,12,13,14,15], despite the improvement of supportive care in sepsis [16]. Among several sepsis mechanisms, the responses against lipopolysaccharide (LPS) during sepsis are extensively studied [17,18,19,20]. The presence of LPS, a major cell wall component of Gram-negative bacteria in the blood (endotoxemia) during sepsis, may be due to Gram-negative bacteremia or the translocation of LPS from the gut because of the high abundance of Gram-negative bacteria in the gut [21,22,23]. Because monocytes and macrophages are important for controlling microbial molecules, both acute responses to LPS and the reduced cytokine production after the first LPS stimulation (LPS tolerance), are possible during prolonged endotoxemia [24,25]. The causes of LPS tolerance in macrophages might be due to epigenetic modifications, chromatin remodeling, and interferences on cell energy status [26,27,28]. Following LPS stimulation, epigenetics (phenotypic alterations without changes in the DNA sequence [29,30]) are responsible for the switch-on and switch-off of DNA transcription through DNA methylation (methyl group added onto the DNA), histone modifications, and noncoding RNA (microRNA) [2]. Of these, the DNA methylation and histone modification are mediated by three groups of enzymes, including (i) the writers (methylation, acetylation, phosphorylation, and ubiquitination), (ii) the erasers (removal of the modifications), and (iii) the readers (binding to different covalent modifications by the writers to mediate physiological outcomes) [3]. Within DNA methylation, the epigenetic regulation of DNA repair is profoundly interesting.

Any forms of electrophilic species (oxidative stress) from several cellular processes, including regular cell activities (stress, cell adaptation, tissue integrity, and remodeling to adapt to the microenvironment), metabolic activation (such as LPS stimulation), and exposure to alkylating agents, induce DNA methylation, especially the purine N -methylation and O6-methylguanine (O6MeG) [31]. Some of these methylation events, particularly O6MeG, are common processes of DNA damage that can trigger point mutations with high mutagenicity and carcinogenicity [32]. Because polymerase enzymes frequently mis insert thymine instead of cytosine (O6MeG:T mismatch) due to the roughly equal strength of the hydrogen bonds to cytosine and thymine; O6MeG, which is commonly activated by the alkylating agent, is mutagenic [33]. However, O6MeG is produced not only by alkylating agents and environmental compounds but also by several endogenous factors during regular cell activities, especially oxidative stress [34]. Indeed, normal metabolic processes, such as hydrolysis, deamination, alkylation, and oxidation, result in several forms of DNA damage, including base damage, single-strand breaks (SSB), double-strand breaks (DSB), and inter-strand cross-links with roughly more than 50,000 lesions per cell daily (approximately 30,000 nucleoside sites in DNA per cell [35]). Although modifications at the O6 position of guanine (O6MeG) might be not high compared with the total number of DNA lesions [36], an abundance of O6MeG might originate from high oxidative stress during sepsis or LPS response. Indeed, chronic inflammation can induce both cancer mutation and DNA methylation [37]. Because of the easy point mutation of O6MeG, O6MeG might cause more severe DNA damage and cell death compared to other types of DNA methylation [38].

To maintain genome stability, DNA repair is necessary, partly through the removal of methyl groups on the DNA by base excision repair initiated by the alkyladenine-DNA glycosylase, the family of alkylation B (AlkB) homolog proteins, and the suicidal enzyme O6-methylguanine-DNA methyltransferase (MGMT) [32,39]. The alkylated and methylated forms of these enzymes are rapidly degraded after DNA repair. Indeed, the intact MGMT in several tissues prevents malignant transformation, and MGMT blockage is used for adjuvant chemotherapy [32] through the blockage of DNA repair in cancers that depend on the rate of MGMT re-synthesis of the malignant cells [39]. Due to several macrophage activator molecules and immune activation-induced oxidative stress in macrophages [40], DNA methylation and DNA damage in the activated cell are possible, especially after stimulation with LPS. Indeed, LPS causes DNA methylation/damage in macrophages directly [41] and indirectly through Toll-like receptor activation and reactive oxygen species (ROS), respectively [42]. Additionally, the higher levels of genomic DNA methylation patterns and hypermethylated genes associated with the pro-inflammatory pathways are demonstrated in patients with sepsis [43]. Despite a solid body of data on the role of O6MeG in malignant cells, the impact of O6MeG and MGMT on macrophages which are the cells with possibly high stress-induced DNA methylation [44] are still unclear. Because (i) sepsis and LPS induce DNA methylation, including Q6MeG [43], (ii) MGMT reduces O6MeG during the DNA repair, and (iii) an MGMT inhibitor (Lomeguatrib) enhances the death of cancer cells (DNA repair interference) [45] that might be beneficial in sepsis [46], the blockage of MGMT in macrophages might reduce macrophage activity and attenuate sepsis-related hyper-inflammation. Furthermore, epigenetic inhibitor screening has also shown that MGMT inhibitors alter the expression of inflammatory cytokine in the LPS-activated macrophages [47] and the MGMT inhibitors not only neutralize O6MeG in DNA but also link to the repair of other pathways [48]. Despite several ongoing research topics, the control of macrophage responses through their manipulations on epigenetics is interesting for controlling immune responses during sepsis [13,49].

Here, we explored the impact of *mgmt* on the responses to LPS, both for a single LPS activation (hyper-inflammatory responses) and double stimulation (LPS tolerance), in vitro and in vivo, using the conditional *mgmt* deletion mice with LysM-Cre system that selectively affects *mgmt* only in macrophages. 

## 2. Results

### 2.1. Transcriptomic Analysis of the Influence of Epigenetic Alteration in Macrophages after Activation by Single or Double LPS Stimulation 

The difference between control cells versus activated macrophages from wild-type (WT) mice using a single LPS stimulation (N/LPS) and LPS tolerance (LPS/LPS) was evaluated by RNA sequencing analysis (Figure 1A). There were 2775 and 3934 up- and down-regulated genes in the LPS-activated macrophages compared with the media control, as indicated by a Volcano plot analysis (Figure 1B). Meanwhile, there were 2115 and 2729 up- and down-regulated genes in the macrophages with LPS tolerance compared to the control (Figure 1C). The differences among the control, single LPS dose, and LPS tolerance conditions were clearly demonstrated by the heat map graphic pattern for DNA and histone modification (Figure 1D,E). Notably, only the genes with significant differences or a tendency of difference among these groups were included. For DNA modification, the genes between control and LPS stimulation, *dnmt1* (DNA methyltransferases) and *mgmt* (O6-methylguanine-DNA methyltransferase), were lower, while *mbd2* (methyl-cytosine binding domain2), *sap30* (Shrimp alkaline phosphatase30), *mecp2* (methyl-CpG binding protein2), and *hdac1* (histone deacetylase) were higher compared with the control (Figure 1D). Additionally, *hdac1* was categorized in both DNA and histone modification due to the collaboration of *hdac1* with *dnmt1* to form a complex [50]. Interestingly, the expression of these DNA modification genes in macrophages between single LPS and LPS tolerance was opposite, as indicated by the heat map analysis pattern (Figure 1D). For the histone modification genes between control and LPS stimulation, *ezh1* (histone-lysine N-methyltransferase1) and *aurkb* (aurora kinase B) were lower than control (Figure 1E), while *sirt6* (sirtuin6), *kmt5a* (lysine methyltransferase 5a), *rnf2* (ring finger protein 2), *hdac1*, and *kdm6b* (lysine demethylase 6b) were higher than the control (Figure 1E). In addition, the expression of these histone modification genes in macrophages between single LPS and LPS tolerance was also the opposite, as indicated by the heat map analysis, except for *sirt6* and *kmt5a* (Figure 1E). Hence, these data supported the idea of an epigenetic alteration in the macrophage responses against LPS, which might be different between single and double LPS stimulations.

### 2.2. Reduction in Macrophage Pro-Inflammation with mgmt Interferences after Either a Single LPS or Two Sequential LPS Stimulations: The Beneficial Effect in Hyper-Inflammatory Sepsis, but Not LPS Tolerance 

The relationship between *mgmt* and the LPS responses was initially explored through silencing of the *mgmt* gene by siRNA (*mgmt* siRNA) in the macrophage cell line (RAW264.7) (Figure 2A). The *mgmt* silencing in either LPS-stimulated macrophages (N/LPS) or LPS tolerance (LPS/LPS) reduced the secretion of pro-inflammatory cytokines (TNF-α and IL-6, but not IL-10) and M1 pro-inflammatory polarization, as indicated by inducible nitric oxide synthase (*iNOS*) and interleukin-1β (*IL-1β*) (Figure 2B–F), the upregulated M2 anti-inflammatory polarization marker arginase-1 (*Arg-1*), and the transforming growth factor-β (*TGF-β*), but not resistin-like molecule-1 (*Fizz-1*), compared with the littermate control cells (mgmt control) (Figure 2G–I). Additionally, there was a reduction in secreted cytokines (TNF-α and IL-6) in the LPS/LPS macrophages with both *mgmt* non-siRNA control and *mgmt* siRNA when compared to a single LPS stimulation (N/LPS), supporting the main characteristics of LPS tolerance as previously described [24,51,52], with the higher level of these cytokines in *mgmt* non-siRNA over the *mgmt* siRNA cells (Figure 2B,C). These data imply reduced macrophage cytokine production, perhaps due to the methylation of the DNA that is responsible for these cytokines, or due to a lack of MGMT to repair the DNA [44]. Moreover, *mgmt* silencing also reduced the pro-inflammatory M1 polarization (*iNOS* and *IL-1β*) and enhanced the M2 anti-inflammatory polarization (*Arg-1* and *TGF-β*) of macrophages compared with the non-silencing cells (Figure 2E–H). Notably, prominent M1 polarization (*iNOS* and *IL-1β*) with lower M2 polarization (*Arg-1* and *TGF-β*) in single LPS stimulation over LPS tolerance, and profound M2 polarization (*Arg-1*) with low M1 polarization (*iNOS* and *IL-1β*) in LPS tolerance over single LPS activation, in either non-siRNA or *mgmt* siRNA groups, were also demonstrated (Figure 2E–H). To further investigate the *mgmt* impacts on LPS stimulation, macrophages from the mice with the conditional *mgmt* deletion (the LysM-Cre system) were then used (Figure 3A). Similar to the *mgmt*-silencing siRNA, the *mgmt* null macrophages (*mgmt*^fl/fl^; LysM-Cre^cre/−^) displayed lower supernatant cytokines (TNF-α, IL-6, and IL-10), down-regulated cytokine genes, and up-regulated M1 polarization (*iNOS* and *IL-1β*), without the alteration of M2 polarization genes, when compared with the littermate cells (*mgmt*^fl/fl^; LysM-Cre^−/−^) (*mgmt* control) (Figure 3B–G). Additionally, lower cytokine production in LPS/LPS compared with N/LPS (the characteristics of LPS tolerance) was evident in macrophages from both mouse strains (*mgmt* control and *mgmt* null) (Figure 3B–G). However, most of the cytokines (supernatant and gene expression) in LPS/LPS *mgmt* null macrophages were lower than the LPS/LPS in cells from littermate mice (*mgmt* control) (Figure 3B–G). In addition, there were less prominent M1 pro-inflammatory polarization genes (*iNOS* and *IL-1β*) with higher M2 polarization (*Arg-1* but not *TGF-β* and *Fizz*) (Figure 3H–L), implying a possible more severe LPS tolerance (low cytokine production and high M2 anti-inflammatory direction that might be inadequate for the microbial control) in *mgmt* null macrophages than in the control. These data suggest that MGMT blockage might be beneficial for the anti-inflammation but can worsen LPS tolerance through the overwhelming anti-inflammatory response direction of the macrophages.

Despite the down-regulation of *mgmt* in 24 h LPS-stimulated macrophages in the RNA sequencing analysis (Figure 1D), macrophages from *mgmt* null mice were further tested (Figure 4A) and the *mgmt* expression in single LPS (N/LPS) was similar to the control (Figure 4B). Meanwhile, *mgmt* was upregulated in LPS/LPS, compared with the control groups, in both the RNA sequencing (Figure 1D) and PCR analyses (Figure 4B). These data imply the obvious need for MGMT enzyme as a DNA repair factor in LPS tolerance, and possibly less need for it in single LPS activation. Because reactive oxygen species (ROS) are natural products during regular processes of the cells, especially by mitochondrial activation, and ROS-induced DNA methylation is known, the non-difference *mgmt* expression in macrophages between control and 24 h LPS might have been due to the similar MGMT levels in both conditions, and the reduced MGMT might have affected the macrophages. Indeed, the *mgmt* null macrophages were more vulnerable to LPS-induced injury, as demonstrated by the reduction of cell viability (MTT assay) and the increased ROS of *mgmt* null cells in either single or double LPS stimulation compared with the *mgmt* control cells (Figure 4C,D), despite the neutral *mgmt* expression of LPS-stimulated cells versus control. Interestingly, the ROS in the *mgmt* null cells with N/LPS was higher than in the LPS/LPS cells (Figure 4D), perhaps due to the inadequate MGMT in N/LPS compared with the LPS/LPS group. In parallel, there was an increase in cell proliferation after LPS stimulation with either the single or double LPS protocols (more prominent in LPS tolerance) in *mgmt* control cells but not in *mgmt* null macrophages (Figure 4B). Meanwhile, the ROS level after single LPS stimulation was not different from the LPS tolerance in both strains of macrophages (Figure 4D), implying a lack of difference in ROS production between the single versus twice LPS stimulations. Additionally, the possible cell damage after LPS activation and LPS tolerance was demonstrated by an elevation of supernatant cell-free DNA compared with the control group, which was similar between N/LPS and LPS/LPS (Figure 4E). Additionally, there was a DNA break, as was indicated by the immunofluorescent staining of phosphohistone H2A.X after the single and twice LPS activations in both *mgmt* control and *mgmt* null macrophages with a similar intensity between groups (Figure 4F,G). However, the highest intensity of phosphohistone H2A.X in the nuclei of the *mgmt* null cells after LPS/LPS activation highlighted the positive DNA damage in the *mgmt* null macrophages with LPS tolerance (Figure 4G). Meanwhile, the positive green color in the other groups was not in the nuclei, which, for at least some of them, might have been a false positive result (Figure 4G). Due to the association between the cell energy status versus cell activities [23,53,54,55,56], the extracellular flux analysis between *mgmt* null and the control was examined (Figure 4H–K). There was a reduction in maximal respiratory capacity (mitochondrial activity), without glycolysis alteration, similarly in both strains of macrophages after activation by either one or two doses of LPS (Figure 4H–K). Thus, the *mgmt* gene seemed to have less impact on the cell energy status, despite the evident impact on cytokine production.

### 2.3. The mgmt Null Mice Demonstrated Less Pro-Inflammatory Cytokine Production in Both Single LPS Injection and LPS Tolerance 

Because of the characteristics of *mgmt*-manipulated macrophages (Figure 3 and Figure 4), further experiments with *mgmt* littermate control (*mgmt*^fl/fl^; LysM-Cre^−/−^) and *mgmt* null mice (*mgmt*^fl/fl^; LysM-Cre^cre/−^) were performed using a single LPS injection (N/LPS) and LPS tolerance (LPS/LPS) (Figure 5A). After a single LPS injection, serum cytokines (TNF-α, IL-6, but not IL-10) in the *mgmt* null mice were lower than in the LPS-administered *mgmt* control (Figure 5B–D), similar to the lower secreted cytokines in LPS-activated *mgmt* null macrophages compared to the control cells (Figure 3B–G). In LPS tolerance, the characteristics of lower serum cytokines in double LPS stimulation compared with only one LPS injection were demonstrated by all of these cytokines in the *mgmt* littermate control mice (open circles versus open square in Figure 5B–D). Meanwhile, this feature was demonstrated only by serum TNF-α (1 h of the protocol) and IL-10 (1 and 3 h of the protocol) in the *mgmt* null mice (blue circle versus red square in Figure 5B–D). Within the LPS/LPS stimulation (LPS tolerance), only serum IL-10 was lower in the *mgmt* null mice, with similar levels of other cytokines (TNF-α and IL-6) compared with the LPS/LPS *mgmt* control mice (open square versus red square in Figure 5B–D), suggesting a similar feature of LPS tolerance between these mouse strains. 

## 3. Discussion

### 3.1. Epigenetic Regulation of Macrophage Responses to LPS Is an Interesting Strategy in Immune Response Manipulation for Sepsis 

Endotoxemia, the presence of lipopolysaccharide (LPS) in blood, can be found in several conditions, including sepsis, uremia, and obesity [57,58,59], mainly because of gut barrier damage [1,22,24,60] and Gram-negative bacteremia [1,61,62]. As one of the pathogen-associated molecular patterns (PAMPs), LPS stimulates all cells in the body, including macrophages, which are the major innate immune cells responsible for the recognition of foreign molecules [53,63]. In the macrophages, the responses to LPS with both a single LPS stimulation and LPS tolerance induced both DNA and histone modifications, as indicated by the RNA sequencing analysis (Figure 1D,E). Indeed, DNA methylation and acetylation at either the cytosine-phosphate-guanine (CpG) or non-CpG sites, and the histone alteration at the N-terminal tails, were critical regulators of gene expression through the chromatin structures [64]. Several key enzymes of epigenetic-induced DNA modification (methylation and acetylation) at the DNA promoter regions controlled the chromatin accessibility, which is well-known in cancer [65]; however, fewer data exist for sepsis [66]. Generally, DNA methylation is mostly the transfer of a methyl group to the C-5 position of the cytosine ring of DNA by DNA methyltransferase (DNMT) onto any cytosines of the genome, especially at the CpG regions [67]. Indeed, DNMT1 induces DNA methylation in macrophages, enhancing pro-inflammatory direction as the deletion of *dnmt1* enhances anti-inflammatory macrophages [44]. Hence, the depletion of *dnmt1* and methyl-binding domain3 (*mbd3*; the linkage of histone methylation to the regions of DNA methylation) [68] in the LPS-stimulated macrophages compared with the control might have been an adaptation to reduce the LPS-induced hyper-inflammation. In contrast, the enhanced *dnmt1* in the LPS/LPS macrophages compared with the LPS macrophages (Figure 1D) might have served to increase the pro-inflammatory direction during the too-low cytokine production of LPS tolerance. Although methylation at the cytosine of CpG regions is common [69], methylation of guanine at the N-7 and O-6 positions of guanine, referred to as N7MeG (or 7-MG) and O6MeG, respectively, that are controlled by *mettl1* (Methyltransferase 1, tRNA methyl-guanosine) and *mgmt* (O6-methylguanine-DNA methyltransferase), respectively, have also been described, especially for cancers and cell viability [70,71]. While *mettl1* was not in the significant genes list from macrophages in this study, alteration of *mgmt* was observed in both the single and double LPS stimulations (Figure 1D). Despite very scarce data on MGMT in macrophages, MGMT in cancer cells has been characterized as an enzyme that removes the methyl group from O6MeG, which enhances the cell viability, and too little MGMT might cause cell injury from the blockage of DNA translation by O6MeG on the DNA [72]. Similar to *dnmt1*, the low *mgmt* at 24 h compared with the control (Figure 1D) might have been a self-adaptation of the LPS-activated macrophages to reduce pro-inflammatory cytokines through sustained DNA damage. However, DNA damage after LPS stimulation is possibly insufficient to reduce cytokine production. Meanwhile, increased *mgmt* in LPS/LPS macrophages compared with the LPS alone (Figure 1D) might have been aimed at neutralizing DNA damage to increase cytokine production, thus counteracting LPS tolerance. Similarly, increased *mgmt* in LPS tolerance seemed to be inadequate as cytokine production was still low, despite the possible reduction of DNA damage. Notably, *mgmt* was the only enzyme that was responsible for guanine methylation from the list of RNA sequencing analyses here. For histone modification in LPS-activated macrophages [73], some of the enzymes were different among the single LPS, LPS tolerance, and control groups. For example, *ezh1* (Enhancer of zeste homolog1), an enzyme for the methylation of lysine on histones (H3K27Me) that inhibits the DNA reading [74], was down-regulated only in the single LPS stimulations, perhaps causing the hyper-responsiveness against LPS. In addition, *rnf2* (Ring finger 2; the core subunit of polycomb repressor complex 1) was up-regulated only in LPS alone, perhaps to control hyper-inflammation [75], and the *hdac1* (histone deacetylases1) [76] and *kdm6b* (lysine demethylase 6B) [77] enzymes responsible for the removal of acetyl and methyl groups from lysine, respectively, were up-regulated both in LPS alone and LPS tolerance (Figure 1E). 

These data support the effect of DNA and histone modification in macrophages during LPS activation [78]. Because of (i) previous reports of DNA methylation (*dnmt*) and histone modification (*ezh*) on sepsis [43,73,79,80], (ii) possible enhanced cell injury from the presence of O6MeG due to the loss of *mgmt* for DNA repair [72], (iii) the availability of the MGMT inhibitor for oncotherapy [81] and its possible use in some chemotherapeutic strategies as a sepsis immune modulator [82], and (iv) epigenetic changes in LPS-activated macrophages and the reversal of LPS tolerance by the *mgmt* inhibitors [47,83], further tests on *mgmt* will be interesting. Theoretically, MGMT can cause less DNA methylation, leading to better cytokine production, which might correlate with more severe inflammation in single LPS and be beneficial in LPS tolerance. In contrast, MGMT blockage might reduce cytokines through enhanced DNA damage, which might be beneficial in hyper-inflammatory sepsis but harmful to LPS tolerance. Indeed, too few cytokines might be inadequate for proper inflammation for the microbial control process, thus leading to secondary infection [84,85,86]. 

### 3.2. Influence of mgmt Enzyme on Sepsis-Related Hyper-Inflammation and Immune Exhaustion 

To test the effects of *mgmt* in sepsis, *mgmt* silencing by siRNA in RAW264.7 cells (a macrophage cell line) and bone marrow-derived macrophages from *mgmt* null mice (*mgmt^fl/fl^; LysM-Cre^cre/−^*) were used. Both siRNA-deleted *mgmt* and *mgmt* null macrophages displayed anti-inflammatory effects as supernatant cytokines (TNF-α and IL-6), and genes of M1 pro-inflammatory macrophages (*iNOS* and *IL-1β*) in both *mgmt*-deleted cells with either a single LPS or LPS tolerance were similarly lower than the control (Figure 2 and Figure 3). There was a prominent anti-inflammatory state of LPS tolerance over the LPS alone in the control macrophages, as indicated by the lower cytokine responses (TNF-α and IL-6) and higher *Arg-1* (an M2 macrophage polarization gene). However, *mgmt* deletion further directed LPS tolerance macrophages into a more prominent anti-inflammatory state as there were even lower inflammatory cytokines (TNF-α) (Figure 2B and Figure 3B) and higher *Arg-1* (Figure 2G and Figure 3J) compared with LPS tolerance in control cells. These data suggest a possible anti-inflammatory effect of *mgmt* blockage; however, this might be harmful in LPS tolerance or sepsis-induced immune exhaustion. In mice, the characteristics of LPS tolerance, as indicated by lower serum cytokines in the double LPS injection compared with a single LPS administration, were observed in both littermate control (*mgmt^fl/fl^; LysM-Cre^−/−^*) and *mgmt* null (*mgmt^fl/fl^; LysM-Cre^cre/−^*) mice.

Due to the influence of *mgmt* on DNA repair through the removal of O6MeG (methyl group at the sixth oxygen molecule on guanine) of DNA, the effects of *mgmt* deletion in both macrophages and mice, together with the altered *mgmt* in RNA sequencing analysis, indicate the possible DNA methylation in activated macrophages (Figure 1). Additionally, macrophage injury, including increased reactive oxygen species (ROS) and cell death after LPS-induced activation, might be partly due to DNA damage [87,88]. Mice with acute endotoxemia [21,89] or with chronic LPS elevation (a possible LPS tolerance) displayed increased spleen apoptosis [56] similar to the in vitro apoptosis of immune cells after LPS stimulation [90]. Although dihydroethidium (DHE; a representative ROS) and mitochondrial injury (reduced maximal respiratory capacity) was worse in the *mgmt* null and control macrophages after both types of activation (single and double LPS), DHE was more profound in the *mgmt* null groups, especially with LPS tolerance (Figure 4). Additionally, the possible higher cell injury in the *mgmt* null macrophages over the control also manifested through lower cell viability (MTT assay) in the *mgmt* null macrophages after both the single and double LPS stimulations. From the MTT assay, both LPS and LPS tolerance stimulated macrophages and increased cell proliferation in the control macrophages (more prominent in LPS tolerance) but not in the *mgmt* null macrophages, possibly due to the higher cell injury. Additionally, the higher supernatant cell-free DNA in activated *mgmt* null macrophages than the control, in either single or twice LPS stimulation, supported a possible susceptibility to LPS-induced cell injury in macrophages without MGMT enzyme. Because phosphorylation of the histone variant H2A.X is a key factor for DNA damage response to assembly of the DNA repair proteins at the chromatin damaged sites, immunofluorescence staining of phosphohistone H2A.X is frequently used to detect DNA damage [91]. Despite the detectable phosphohistone H2A.X in all groups of macrophages (control, N/LPS, and LPS/LPS), the intensity of the damage in N/LPS and LPS/LPS was higher than the control, indicating possible LPS-induced DNA damages that were similar in the *mgmt* null and control cells (Figure 4F,G). Although the loss of MGMT enzyme did not induce higher DNA damage, as determined by the intensity of the fluorescent color per cell, the high color intensity that was clearly located in the nuclei (the actual site of DNA repair) was demonstrated only in LPS/LPS *mgmt* null macrophages (Figure 4G). Because the positive phosphohistone H2A.X staining outside the nuclei might have been false positive fluorescence, other methods for DNA damage detection are needed for a solid conclusion. 

Nevertheless, the maintenance of cell viability through DNA methylation at guanine seems to be important in the activated macrophages, despite more common methylation at the cytosine residues [67]. Although direct evaluation of O6MeG in macrophages after LPS or LPS tolerance was not performed here, the differences between *mgmt* null macrophages versus control after activation indirectly support the presence of O6MeG in macrophages. It is interesting to note that the standard method for the direct detection of O6MeG is still unclear [92,93], and the detection of a DNA break in immune cells might be different from the well-known protocol for DNA break detection of the parenchymal cells that is mostly studied in cancer topics [94]. In comparison with the control, the difference in cell viability, increased ROS, and elevated cell-free DNA, together with similar reduced cytokine production in stimulated macrophages with a lack of MGMT using *mgmt* null cells, indirectly support an influence of MGMT in LPS-activated macrophages. Although more experiments are required for an in-depth understanding of the mechanisms involved, our initial results support a possible extended use of an MGMT inhibitor (Lomeguatrib), a chemotherapeutic agent [95], on the attenuation of sepsis hyper-inflammation. 

Furthermore, our data also support the idea that serum cytokines, in response to LPS injection (both a single or a double injection), are mainly produced from macrophages (Figure 5B–D), as previously reported [96], and that the signaling blockage only in macrophages, but not in other cells, might be an effective treatment with limited side effects. Due to lower levels of pro-inflammatory cytokines after the first dose of LPS injection, the severity of LPS tolerance, as indicated by the difference between the first and second dose of LPS in *mgmt* null mice, was lower than in the control mice. While the lower serum TNF-α was very obvious in the control mice with LPS/LPS compared with N/LPS, as expected from LPS tolerance, serum TNF-α in LPS/LPS *mgmt* null mice were not different from *mgmt* null mice with LPS alone. Although the *mgmt* depletion in the macrophages protected the mice from too high pro-inflammatory septic shock, the loss of *mgmt* induced too little pro-inflammatory cytokines with more prominent LPS tolerance, which could correlate with the enhanced susceptibility to secondary infection [97]. 

### 3.3. Clinical Aspects and Future Experiments

We hypothesized that activation by a single LPS dose or LPS tolerance would induce injury in macrophages, partly through DNA methylation and acetylation, which need enzymes, such as MGMT, to remove the methyl or acetyl groups in order to revitalize macrophages and maintain the regular cell functions (Figure 6). To continuously produce cytokines, the DNA methylation needed to be repaired and the failure of DNA repair caused a reduction in macrophage function, especially cytokine production (Figure 6). Thus, MGMT inhibition might be useful to attenuate sepsis-related hyperinflammation. Currently, the treatment of some cancers starts with an alkylating agent to induce DNA methylation at guanine (O6MeG), especially Temozolomide, and the combination with Lomeguatrib (an MGMT inhibitor) results in more cancer cell death, taking advantage of the high mutagenicity of O6MeG [95]. In sepsis, MGMT inhibitors might induce injury in the regular cells and short-term administration, and/or the MGMT blockage specifically focused on macrophages might be theoretically better. Notably, good microbial control, especially by effective antibiotics, is necessary for all of the immune-mediated adjunctive therapies in sepsis, and determination of the immune directions is possibly needed. Theoretically, sepsis immune responses are crudely divided into hyper-inflammation and immune exhaustion by several biomarkers. For example, high serum IL-6 and IL-1 might be biomarkers for sepsis-related hyper-inflammation [98,99], while down-regulated HLA-DR and viral reactivation (cytomegalovirus; the common dormant virus in the human host) possibly indicate sepsis-related immune exhaustion [100,101]. Thus, immune monitoring in sepsis is needed for the use of MGMT inhibitors, and down-regulated HLA-DR and lower inflammatory cytokines might be a contraindication because the overwhelming inhibition might escalate infection susceptibility. Because LPS response and LPS tolerance are only a subset of sepsis-related immune responses [20], further evaluations of the effect of MGMT in sepsis are warranted. 

Finally, there are several limitations in our study that should be mentioned. First, supportive information on the transcriptome results, especially from the Western blot analysis, was not performed. Second, the impacts of the MGMT overexpression in macrophages with LPS stimulation were not examined. Third, the direct detection of Q6MeG on macrophages was not conducted here, partly due to the unclear standard proper methods of O6MeG detection [92,93]. Fourth, additional DNA damage detection methods are necessary for the determination of DNA damage. Nevertheless, an initial proof of concept on the impacts of MGMT enzyme in LPS-stimulated macrophages is presented here, which indicates several more interesting experiments on the topic. Overall, we conclude that MGMT inhibitors, an available adjunctive therapy for malignancy, might be beneficial in some situations of sepsis. More studies are warranted.

## 4. Materials and Methods 

### 4.1. Small Interfering RNA (siRNA) in the Macrophage Cell Line

Murine macrophage-like cells (RAW264.7; TIB-71) (American Type Culture Collection; ATCC, Manassas, VA, USA) were maintained in Dulbecco’s Modified Eagle’s Medium (DMEM) supplemented with 10% Fetal Bovine Serum (FBS) in a humidified incubator at 37 °C with 5% CO_2_. The small interfering RNA (siRNA) silencing of *mgmt* was then performed using RAW264.7 at 10^6^ cells/mL seeded into 6-well plates with the siRNA for *mgmt* (Dharmacon^TM^ Accell^TM^, Horizon Discovery, Watwebeach, UK) in siRNA buffer, following a previous publication [73]. Briefly, the siRNA at 1 μM per well was incubated at 37 °C with 5% CO_2_ for 48 h. The non-targeting pool siRNA (Dharmacon^TM^) was used as a control. Macrophages with *mgmt*-siRNA and non-siRNA (non-targeting pool siRNA) were activated by three different protocols for a proper comparison. First, the single lipopolysaccharide (LPS) stimulation protocol began with DMEM, followed by LPS (*Escherichia coli* 026: B6; Sigma-Aldrich, Waltham, MA, USA) (100 ng/mL) 24 h later (N/LPS). Second, the LPS tolerance protocol used two sequential stimulations of 100 ng/mL LPS at 24 h and 48 h (LPS/LPS) in the experiments. Third, the control protocol (N/N) was performed by the incubation in twice-changed DMEM only. After another 24 h of each protocol, the sample collection (supernatant and cells) was then performed and supernatant cytokines (TNF-α, IL-6, and IL-10) were evaluated by ELISA (Invitrogen, Carlsbad, CA, USA). Meanwhile, the gene expression was evaluated by quantitative real-time polymerase chain reaction (PCR), as previously described [102,103,104,105,106]. In brief, the PCR started with RNA extraction from the cells with TRIzol Reagent (Invitrogen), together with RNeasy Mini Kit (Qiagen, Hilden, Germany); then 1 mg of total RNA was used for cDNA synthesis with iScript reverse transcription super-mix (Bio-Rad, Hercules, CA, USA). Quantitative real-time PCR was performed on a QuantStudio 6 real-time PCR system (Thermo Fisher Scientific, San Jose, CA, USA) using SsoAdvance Universal SYBR Green Super-mix (Bio-Rad). The gene expression was normalized to beta-actin (*β-actin*; an endogenous housekeeping gene) and the fold change was calculated by the ∆∆Ct method. The primers used in this study are listed in Table 1.

### 4.2. The Transcriptome Analysis

The RNA sequencing analysis in WT macrophages was performed to determine epigenetic alterations in macrophages with LPS stimulations (a single LPS stimulation and LPS tolerance). Bone marrow-derived macrophages (BMDM) were prepared from the femurs of mice using supplemented Dulbecco’s Modified Eagle’s Medium (DMEM) with a 20% conditioned medium of the L929 cells (ATCC CCL-1), as previously described [53,54,56,106]. Macrophages at 5 × 10^4^ cells/well in supplemented DMEM (Thermo Fisher Scientific) were incubated in 5% carbon dioxide (CO_2_) at 37 °C for 24 h before being treated by 3 experimental protocols, as mentioned above. The RNA from macrophages extracted by a RNeasy mini kit (Qiagen) was then processed with the RNA sequencing (BGISEQ-50) platform, as previously published [107]. The mRNA analysis was conducted based on triplicate macrophage samples. The FastQC was used to determine the sequencing quality. The raw sequencing reads were mapped and aligned against *Mus musculus* reference genome GRCm39 using STAR [108], followed by gene quantification against the reference mouse transcriptome by Kallisto [109]. Read counts were normalized and analyzed (differentially expressed genes; DEGs) using the edgeR [110] and limma-voom packages [111,112]. Genes were considered significant differences (*p*-value < 0.05) when the log2 value of fold change level was less than −2 or greater than 2, indicating down- or up-regulation, respectively. The DEGs clustering was performed based on Euclidean distance and the Ward.D2 method with the ComplexHeatmap package [113], and the log2 expression (TPM; transcript count per million) of selected epigenetic-related genes [114] was compared to determine statistical significance using the Wilcoxon test in the ggpubr package [115]. A *p*-value less than 0.05 indicated statistical significance.

### 4.3. The Intro Experiments 

Bone marrow-derived macrophage from *mgmt* control (*mgmt*^fl/fl^; LysM-Cre^−/−^) or *mgmt* null (*mgmt*^fl/fl^; LysM-Cre^cre/−^) mice were extracted from mouse femurs before activation by several protocols (N/LPS, LPS/LPS, or N/N). The supernatant cytokines (TNF-α, IL-6, and IL-10) and gene expression were measured by ELISA and PCR, as mentioned above. Because of the influence of cell viability and reactive oxygen species (ROS) in cell injury, MTT (3-[4,5-dimethylthiazol-2-yl]-2,5 diphenyl tetrazolium bromide) assay and dihydroethidium (DHE) were measured according to the published protocols [51,58,116]. For MTT, the activated cells were incubated with 0.5 mg/mL of MTT solution (Thermo Fisher Scientific) for 2 h at 37 °C in the dark, before MTT removal with the dilution with dimethyl sulfoxide (DMSO), and measured with a Varioskan Flash microplate reader at an absorbance of optical density at 570 nm. Additionally, DHE (Sigma-Aldrich) at 20 µM was incubated in the activated macrophages for 20 min at 37 °C before DHE measurement, and the fluorescence readings were analyzed at 520 nm by a Varioskan Flash microplate reader and presented by the fluorescence arbitrary unit. Supernatant cell-free DNA was detected by Quanti PicoGreen assay (Sigma-Aldrich). For DNA break determination, macrophages at 3 × 10^6^ cells were seeded on glass-bottomed 6-well plates before activation (N/LPS, LPS/LPS, or N/N). The cells were then fixed with 4% paraformaldehyde in Tris Buffered Saline (TBS) for 15 min, permeabilized with 0.1% triton X-100, and subsequently washed three times in 1X TBS with 0.05% Tween-20. The fixed samples were blocked with 2% bovine serum albumin in 1X TBS for 1 h at room temperature and then incubated overnight at 4 °C with phospho-histone H2A.X (Ser139) (20E3) rabbit mAb (Cell signaling). Proteins were visualized using goat anti-mouse IgG H&L tagged Alexa Flour 488 (Abcam; ab150113) (green) and actin filaments were labeled with DY-554 phalloidin (red); the fluorescent intensity per cell was evaluated by confocal laser scanning microscope with the ZEN 3.0 software (CLSM, Zeiss, Germany) at 630× magnification in 10 randomly selected fields per slide.

### 4.4. Extracellular Flux Analysis

For the cell energy status (extracellular flux analysis), Seahorse XFp Analyzers (Agilent, Santa Clara, CA, USA) were used. As such, the oxygen consumption rate (OCR) and extracellular acidification rate (ECAR) were used to represent mitochondrial function (respiration) and glycolysis activity, respectively, following previous publications [23,55,107,117,118]. In brief, the activated macrophages (1 × 10^5^ cells/well) were incubated in the Seahorse media (DMEM complemented with glucose, pyruvate, and l-glutamine) (Agilent, 103575–100) before activation by different metabolic interference compounds of the protocol, including oligomycin, carbonyl cyanide-4-(trifluoromethoxy)-phenylhydrazone (FCCP), and rotenone/antimycin A for OCR evaluation, or glucose, oligomycin, and 2-Deoxy-d-glucose (2-DG) for ECAR measurement. The maximal respiration and maximal glycolysis capacity were calculated by Seahorse Wave 2.6 software.

### 4.5. Animal and Animal Model

Protocol No. 017/2562 was approved by the Institutional Animal Care and Use Committee of the Faculty of Medicine, Chulalongkorn University, Bangkok, Thailand, according to National Institutes of Health (NIH) criteria. For the transcriptomic analysis, macrophages were prepared from the long bone (femurs and tibias) of 8-week-old male wild-type (WT) C57BL/6 mice purchased from Nomura Siam (Pathumwan, Bangkok, Thailand). In parallel, *mgmt*^flox/flox^ and LyM-Cre^Cre/Cre^ mice were purchased from RIKEN BRC Experimental Animal Division (Ibaraki, Japan) and cross-bred to produce *mgmt* littermate control (*mgmt*^fl/fl^; LysM-Cre^−/−^) or *mgmt* null (*mgmt*^fl/fl^; LysM-Cre^cre/−^) mice in F3 of the breeding protocol. As such, the *mgmt*^flox/flox^ mice with the loxP sites were bred with LysM-Cre^Cre/Cre^ mice. The mice with a Cre recombinase under the control of lysozyme M were used to target *mgmt* for *mgmt* deletion only in the myeloid cells, including macrophages and neutrophils. The offspring were either *mgmt*^flox/flox^ with no LysM-Cre (*mgmt*^fl/fl^; LysM-Cre^−/−^), which were categorized as the littermate controls or *mgmt* control. Meanwhile, the mice that were positive for the Cre driver were *mgmt* null (*mgmt*^fl/fl^; LysM-Cre^cre/−^) mice with a lack of MGMT enzyme. The conditional targeted Cre positive mice (*mgmt* null) were age- and gender-matched with the floxed/floxed littermate controls (*mgmt* control) using the male mice aged 8–10 weeks old. To genotype these mice on the loxP sites insertion, the following primers were used: (i) LysM-cre primer; F: 5′-GAACGCACTGATTTCGACCA-3′, R: 5′-GCTAACCAGCGTTTTCGTTC-3′, (ii) *mgmt*-loxP primer F; 5′-TGGGCTTCAAATCAAGGAACAGAA-3′, R: 5′-AACTATCCTGCTCACTCTCTGTAG-3′, and (iii) Cre recombination (for Cre activity); F: 5′-GGTGTGGATCCCAAGAAATTGAAG-3′, R: 5′-TGTTCAAGAGTGACACACAGTCA-3′ [73]. The mice homozygous for the flox were selected and genotyped for the expression of LysM-Cre using the primers: F: 5′-CTTGGGCTGCCAGAATTCTC-3′, R: 5′-CCCAGAAATGCCAGATTACG-3′. The mice were divided into 3 protocols similar to the in vitro experiments. As such, the LPS tolerance (LPS/LPS) was performed by an intraperitoneal injection of 0.8 mg/kg LPS (*Escherichia coli* 026:B6) (Sigma-Aldrich, St. Louis, MO, USA), with an additional dose of 4 mg/kg LPS 24 h later. In parallel, the single LPS stimulation was conducted by an intraperitoneal injection of phosphate buffer solution (PBS), followed by LPS (4 mg/kg) 24 h later. For the control (N/N) protocol, twice intraperitoneal PBS injection with a 24 h duration between the doses was performed. Following these protocols, blood was collected through tail vein nicking at 1 and 3 h afterward and mice were sacrificed with cardiac puncture under isoflurane anesthesia with blood collection at 6 h after the last injection. Serum cytokines were evaluated by ELISA (Invitrogen). The mouse genotype data are demonstrated in Figure 7.

### 4.6. Statistical Analysis

All the data were analyzed with GraphPad Prism6 and shown as mean ± S.E.M (standard error). One-way analysis of variance (ANOVA) with Tukey’s comparison test was used and a *p*-value less than 0.05 was considered significant. 

## 5. Conclusions

The alteration of several enzymes of epigenetic processes for DNA and histone modifications after single LPS activation and LPS tolerance was demonstrated in wild-type macrophages. Reduced pro-inflammatory cytokines with a single LPS stimulation and more severe LPS tolerance (lower supernatant cytokines with a second dose of LPS) in *mgmt* null *(mgmt^fl/fl^; LysM-Cre^cre/−^)* macrophages and mice compared with the control groups *(mgmt^fl/fl^; LysM-Cre^−/−^)* supported the conclusion regarding the possible benefits and limitations of MGMT blockage in hyper-inflammation and LPS tolerance, respectively. Hence, the use of MGMT blockage which is an available drug in cancer therapy is proposed to attenuate severe inflammation in sepsis. More studies are warranted. 

## Figures and Tables

**Figure 1 ijms-24-10139-f001:**
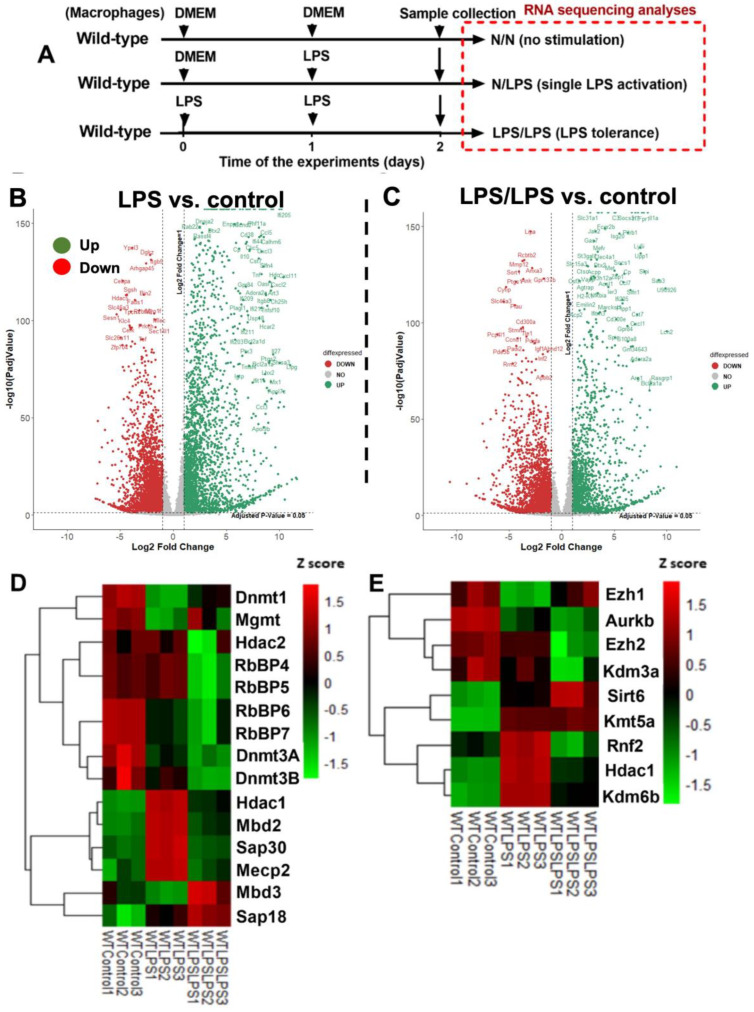
The transcriptome profiles and the log2 of the transcript count per million (TPM) of genes in bone marrow-derived macrophages from wild-type mice activated by lipopolysaccharide (LPS) in a single protocol (N/LPS), which started with the culture media followed by LPS 24 h later or LPS tolerance (LPS/LPS) by the two sequential LPS stimulations, or control (N/N; control) using the culture media incubation only (**A**), as indicated by the Volcano plot analysis (**B**,**C**), the heatmap of the genes of epigenetic changes in DNA methylation or acetylation (**D**), and the histone modification (**E**) with difference (or tendency of difference) to the control. Macrophages were isolated from three different mice. List of abbreviations: *Dnmt* (DNA methyltransferases), *Mgmt* (O6-methylguanine-DNA methyltransferase), *Hdac* (histone deacetylase), *RbBP* (Retinoblastoma-Binding Protein), *Mbd* (methyl-cytosine binding domain), *Sap* (Shrimp alkaline phosphatase), *Mecp* (methyl-CpG binding protein), *Ezh* (histone-lysine N-methyltransferase), *Aurkb* (aurora kinase B), *Kdm* (lysine demethylase), *Sirt* (sirtuin), *Kmt* (lysine methyltransferase), and *Rnf2* (ring finger protein).

**Figure 2 ijms-24-10139-f002:**
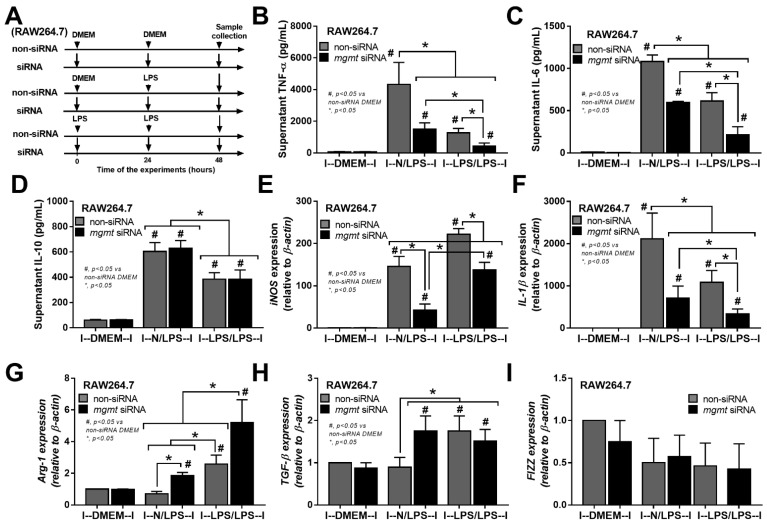
The schema of the experiments in a murine macrophage cell line (RAW264.7) with the silencing of *mgmt* gene using small interfering RNA (*mgmt* siRNA) or the control siRNA (non-targeting pool siRNA; non-siRNA) after activation by lipopolysaccharide (LPS) in a single protocol (N/LPS), which started with the culture media followed by LPS 24 h later or LPS tolerance (LPS/LPS) by the two sequential LPS stimulations, or control (N/N), using the culture media incubation only (**A**). The characteristics of macrophages under these protocols as indicated by secreted cytokines (TNF-α, IL-6, and IL-10) (**B**–**D**), the expression of pro-inflammatory genes of M1 polarization (*iNOS* and *IL-1β*) (**E**,**F**), and the anti-inflammatory genes of M2 polarization (*Arg-1*, *TGF-β*, and *Fizz-1*) (**G**–**I**). Triplicate independent experiments were performed. Mean ± SEM with one-way ANOVA followed by Tukey’s analysis was used. #, *p* ˂ 0.05 *mgmt* vs. control DMEM; *, *p* ˂ 0.05 between the indicated groups.

**Figure 3 ijms-24-10139-f003:**
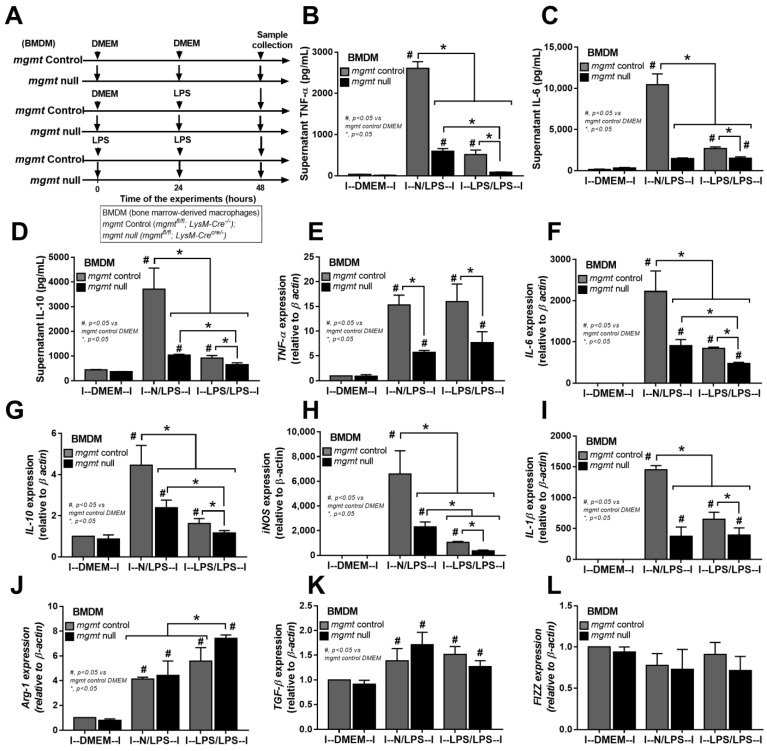
The schema of the experiments in bone marrow-derived macrophages from *mgmt* control (*mgmt*^fl/fl^; LysM-Cre^−/−^) and *mgmt* null (*mgmt^f^*^l/fl^; LysM-Cre^cre/−^) mice after activation by lipopolysaccharide (LPS) in a single protocol (N/LPS), which started with the culture media followed by LPS 24 h later or LPS tolerance (LPS/LPS) by the two sequential LPS stimulations, or control (N/N), using the culture media incubation only (**A**). The characteristics of macrophages under these protocols, as indicated by supernatant cytokines (TNF-α, IL-6, and IL-10) (**B**–**D**), the gene expression of cytokines (*TNF-α*, *IL-6*, and *IL-10*) (**E**–**G**), M1 macrophage polarization (*iNOS* and *IL-1β*) (**H**,**I**), and the M2 macrophage polarization (*Arg-1*, *TGF-β*, and *Fizz-1*) (**J**–**L**), are also demonstrated. Triplicate independent experiments were performed. Mean ± SEM with one-way ANOVA followed by Tukey’s analysis was used. #, *p* ˂ 0.05 *mgmt* vs. control DMEM; *, *p* ˂ 0.05 between the indicated groups.

**Figure 4 ijms-24-10139-f004:**
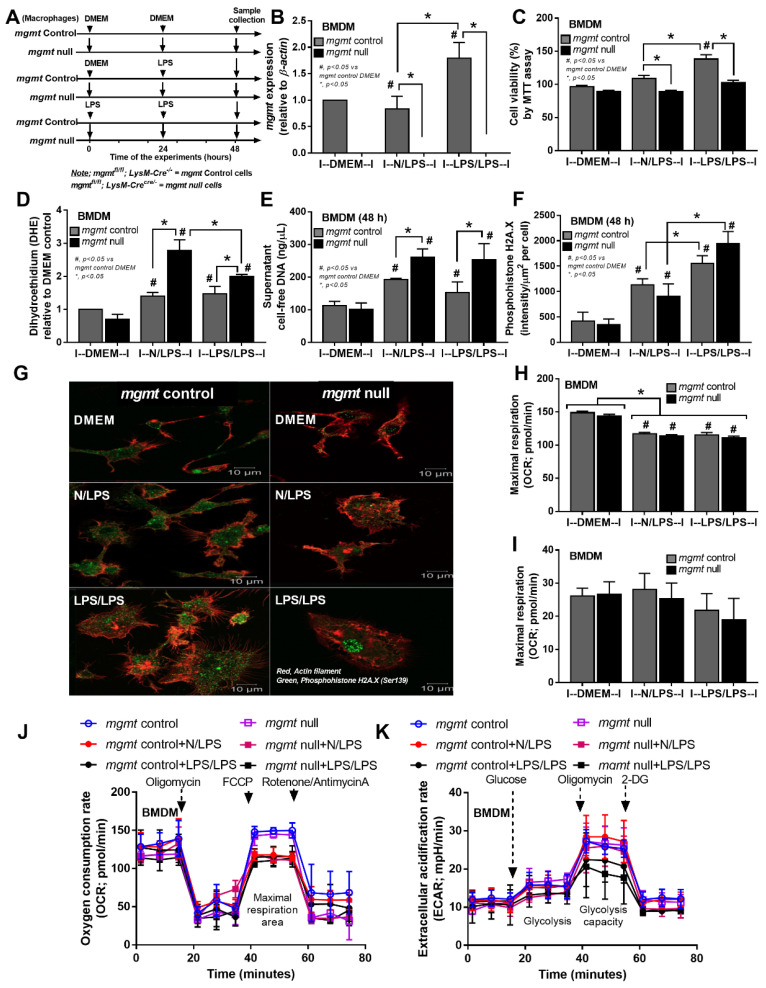
The schema of the experiments in bone marrow-derived macrophages from *mgmt* control (*mgmt*^fl/fl^; LysM-Cre^−/−^ and *mgmt* null (*mgmt*^fl/fl^; LysM-Cre^cre/−^) mice after activation by lipopolysaccharide (LPS) in a single protocol (N/LPS), which started with the culture media followed by LPS 24 h later or LPS tolerance (LPS/LPS) by the two sequential LPS stimulations, or control (N/N), using the culture media incubation only (**A**). The characteristics of macrophages under these protocols, as indicated by the expression of *mgmt* (**B**), cell viability using 3-(4,5-dimethylthiazol-2-yl)-2,5-diphenyl tetrazolium bromide (MTT) (**C**), reactive oxygen species with dihydroethidium stain (DHE) (**D**), supernatant cell-free DNA (**E**), the DNA damage score with the representative immunofluorescent pictures of phosphohistone H2A.X (a DNA break biomarker) (**F**,**G**), and the energy status of cells (extracellular flux analysis) (**H**–**K**). Independent triplicated experiments were performed. Mean ± SEM with one-way ANOVA followed by Tukey’s analysis was used. #, *p* ˂ 0.05 *mgmt* vs. control DMEM; *, *p* ˂ 0.05 between the indicated groups.

**Figure 5 ijms-24-10139-f005:**
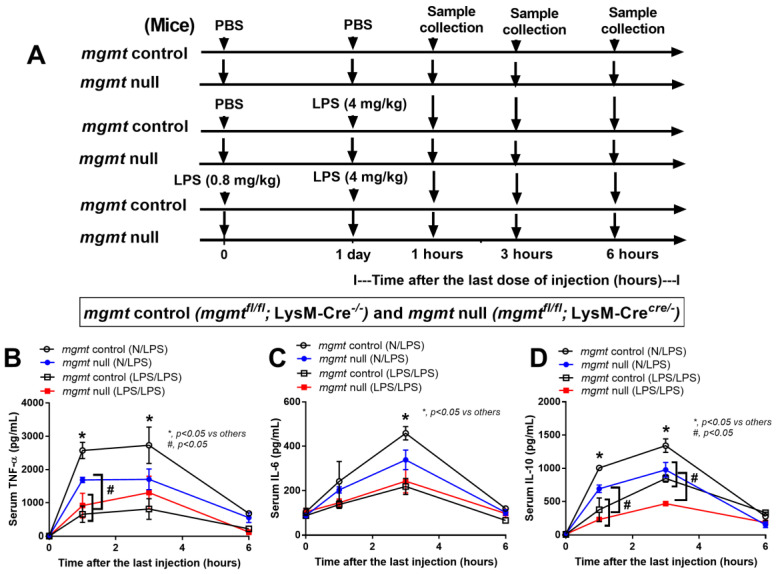
Schematic workflow (**A**) demonstrates the experimental groups, including lipopolysaccharide (LPS) tolerance, started with LPS intraperitoneal (ip) injection (0.8 mg/kg) followed by LPS (4 mg/kg) (LPS/LPS); a single LPS stimulation, started with phosphate buffer solution (PBS) followed by LPS (4 mg/kg) (N/LPS), in *mgmt* control (*mgmt*^fl/fl^; LysM-Cre^−/−^) and *Mgmt* null (*mgmt*^fl/fl^; LysM-Cre^cre/−^) mice as indicated by serum cytokines (TNF-α, IL-6, and IL-10) (**B**–**D**), are demonstrated (*n* = 5–7/group and time-point). Mean ± SEM with one-way ANOVA followed by Tukey’s analysis was used. *, *p* ˂ 0.05 between *mgmt* control versus others; #, *p* ˂ 0.05 between the indicated groups.

**Figure 6 ijms-24-10139-f006:**
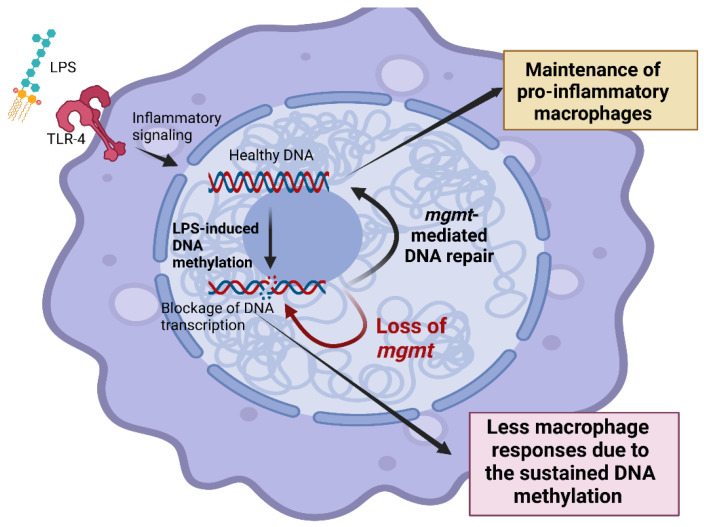
The proposed working hypothesis demonstrated the impact of O6-methylguanine-DNA methyltransferase (MGMT) in responses against lipopolysaccharide (LPS) of macrophages. LPS activates inflammatory responses through Toll-like receptor 4 (TLR-4), which causes methylation in several areas of DNA, including O6-methylguanine (O6MeG). DNA methylation impairs DNA transcription and induces programmed cell death, especially apoptosis [100]. The MGMT enzyme, referred to as “a DNA suicide repair enzyme”, transfers the methyl group at the O6 site of guanine to the cysteine residues of MGMT, allowing macrophages to maintain their functions (yellow-colored box). Without MGMT or using an MGMT inhibitor, there might be an impairment of macrophage cytokine production (red-colored box) that is beneficial in hyper-inflammatory sepsis. This figure was created by BioRender.com.

**Figure 7 ijms-24-10139-f007:**
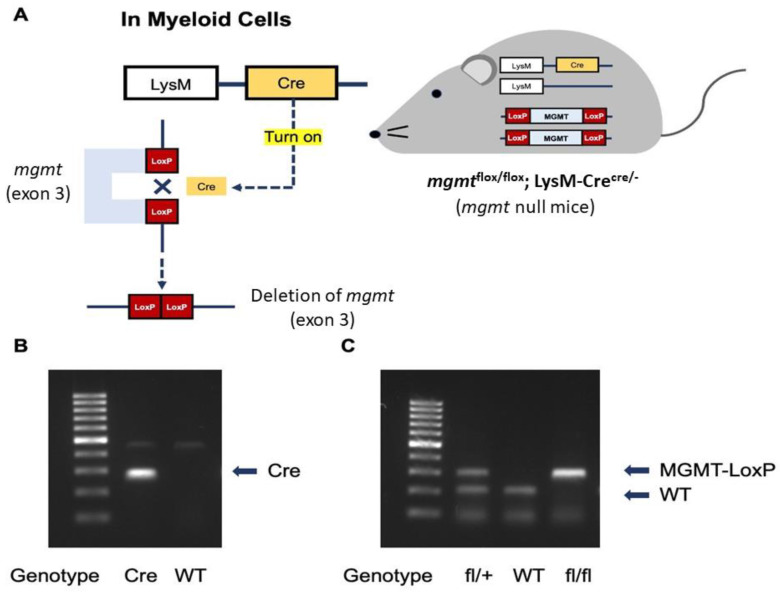
The diagram demonstrates the necessary of Cre to activate LoxP for the deletion of *mgmt* gene (exon 3) only in the myeloid cells of *mgmt* null mice (**A**), and the representative genotype to identify *mgmt* null mice (*mgmt*^flox/flox^; LysM-Cre^cre/−^) with the identified bands of Cre and MGMT-LoxP for the flox/flox (fl/fl) group (**B**,**C**) are demonstrated.

**Table 1 ijms-24-10139-t001:** Lists of primers used in the study.

Name	Forward	Reverse
Tumor necrosis factor α (*TNF-α*)	5′-CCTCACACTCAGATCATCTTCTC-3′	5′-AGATCCATGCCGTTGGCCAG-3′
Interleukin-6 (*IL-6*)	5′-TACCACTTCACAAGTCGGAGGC-3′	5′-CTGCAAGTGCATCATCGTTGTTC-3′
Interleukin-10 (*IL-10*)	5′-GCTCTTACTGACTGGCATGAG-3′	5′-CGCAGCTCTAGGAGCATGTG-3′
Inducible nitric oxide synthase (*iNOS*)	5′-ACCCACATCTGGCAGAATGAG-3′	5′-AGCCATGACCTTTCGCATTAG-3′
Interleukin-1β (*IL-1β*)	5′-GAAATGCCACCTTTTGACAGTG-3′	5′-TGGATGCTCTCATCAGGACAG-3′
Arginase-1 (*Arg-1*)	5′-CTTGGCTTGCTTCGGAACTC-3′	5′-GGAGAAGGCGTTTGCTTAGTT-3′
Resistin-like molecule-α1 (*Fizz-1*)	5′-GCCAGGTCCTGGAACCTTTC-3′	5′-GGAGCAGGGAGATGCAGATGA-3′
Transforming growth factor-β (*TGF-β*)	5′-CAGAGCTGCGCTTGCAGAG-3′	5′-GTCAGCAGCCGGTTACCAAG-3′
O6-methylguanine-DNA methyltransferase (*mgmt*)	5′-CTATTTCCGTGAACCCGCAG-3′	5′-ACCGGATTGCTTCTCATTGC-3′
*β-actin*	5′-CGGTTCCGATGCCCTGAGGCTCTT-3′	5′-CGTCACACTTCATGATGGAATTGA-3′

## Data Availability

Not applicable.

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
