# Peer review of "Less Severe Lipopolysaccharide-Induced Inflammation in Conditional mgmt-Deleted Mice with LysM-Cre System: The Loss of DNA Repair in Macrophages"

_ijms, 2023, doi:10.3390/ijms241210139_

Round 1

Reviewer 1 Report

Saisorn and collaborators present a very well performed and convincing study demonstrating the role of mgmt1 in inflammation.

Notably, the study rests on the generation of mgmt1 lineage specific delation in macrophages and in gene silencing studies in RAW macrophages.

Results are convincing ast the experimental protocols and data presentation have been executed at a very high standard.

I wish to present to the authors one point for imrovement:

I believe that some representative genotypying data shold be presented, either in the main manuscript or as a supplementary figure/material.

Aside from this, it is my opinion that the manuscript should be strongly considered for publication in the journal.

Please just proceed to a well done final proofreading.

Author Response

Reviewer 1

Saisorn and collaborators present a very well performed and convincing study demonstrating the role of mgmt1 in inflammation. Notably, the study rests on the generation of mgmt1 lineage specific deletion in macrophages and in gene silencing studies in RAW macrophages. Results are convincing, the experimental protocols and data presentation have been executed at a very high standard. I wish to present to the authors one point for improvement:

I believe that some representative genotyping data should be presented, either in the main manuscript or as a supplementary figure/material.

ANS: We thank the reviewer for the comment and put the genotyping result in the new figure 7.

Aside from this, it is my opinion that the manuscript should be strongly considered for publication in the journal.

Reviewer 2 Report

This manuscript describes the role of mgmt on the response to LPS in macrophages. The article is interesting and is well-written. However I have several concerns and points that should be improved:

1.- Transcriptome analysis shows differences in several epigenetic related genes. Why authors have been focused on mgmt? Indeed, transcriptomic results should be validated by western bloting analysis.

2.- Authors uses siRNA to silencing mgmt but they can not discart off target effects. Some studies re-expressiong mgmt (after the silencing) would be useful to dismiss of target effects. Indeed, what happens with mgmt overexpression? Does mgmt overexpression increase LPS effects?

3.- In Figure 6 and in the discussion authors propose that LPS induces DNA methylation and that mgmt mediates DNA repair. However authors have not studied DNA neither DNA methylation or DNA damage. It is speculative. Some experiments abouth these pathways would be informative.

4.- In line 22 (mgmt expression in single LPS was similar to the control) the authors reffers to Fig 4A instead 4B.

Author Response

Reviewer 2

This manuscript describes the role of mgmt on the response to LPS in macrophages. The article is interesting and is well-written. However, I have several concerns and points that should be improved:

1.- Transcriptome analysis shows differences in several epigenetic related genes. Why authors have been focused on mgmt? Indeed, transcriptomic results should be validated by western blotting analysis.

ANS: We thank the reviewer for the comment. However, we have a problem on antibody of mgmt, then we put this as a limitation mentioned at the end of the new discussion as following “Finally, there are several limitations in our study that should be mentioned. First, supportive information of the transcriptome results, especially with the Western blot analysis, was not performed. … Nevertheless, a proof of concept on the impacts of MGMT enzyme in LPS-stimulated macrophages was initially presented here which indicates several more interesting experiments on the topic.”.  

2.- Authors uses siRNA to silencing mgmt but they cannot discard off target effects. Some studies re-expressing mgmt (after the silencing) would be useful to dismiss of target effects. Indeed, what happens with mgmt overexpression? Does mgmt overexpression increase LPS effects?

ANS: We thank the reviewer for the comment. The mgmt silencing demonstrated clearly difference from the control group (in parallel to the cells from mgmt-deleted mice) and further repression of MGMT in silencing macrophages might also be in the same direction. For the MGMT over-expression, it is really interesting but, unfortunately, we currently have some problems on mgmt overexpression. Then, we put this as another limitation mentioned at the end of the new discussion.

3.- In Figure 6 and in the discussion, authors propose that LPS induces DNA methylation and that mgmt mediates DNA repair. However, authors have not studied DNA neither DNA methylation or DNA damage. It is speculative. Some experiments about these pathways would be informative.

ANS: We thank the reviewer for the comment and added supernatant cell-free DNA and DNA damage results in the manuscript (new figure 4; using phosphohistone H2A.X immunofluorescent staining).

4.- In line 22 (mgmt expression in single LPS was similar to the control) the authors refer to Fig 4A instead 4B.

ANS: We thank the reviewer for the comment and correct it accordingly.

Round 2

Reviewer 1 Report

-

Reviewer 2 Report

Some of my concerns have been adressed and manuscript might be accepted